# High-throughput phenotyping analysis of maize at the seedling stage using end-to-end segmentation network

Yinglun Li[1,2], Weiliang Wen[2,3]*, Xinyu Guo[2,3], Zetao Yu[3], Shenghao Gu[2,3], Haipeng Yan[4], Chunjiang Zhao[1,2,3]*

**1** College of Resources and Environment, Jilin Agricultural University, Changchun, China, **2** Beijing Research Center for Information Technology in Agriculture, Beijing, China, **3** Beijing Key Lab of Digital Plant, National Engineering Research Center for Information Technology in Agriculture, Beijing, China, **4** Beijing Shunxin Agricultural Science and Technology Co., Ltd, Beijing, China

* wlwen37@163.com (WW); zhaocj@nercita.org.cn (CZ)

## Abstract

Image processing technologies are available for high-throughput acquisition and analysis of phenotypes for crop populations, which is of great significance for crop growth monitoring, evaluation of seedling condition, and cultivation management. However, existing methods rely on empirical segmentation thresholds, thus can have insufficient accuracy of extracted phenotypes. Taking maize as an example crop, we propose a phenotype extraction approach from top-view images at the seedling stage. An end-to-end segmentation network, named PlantU-net, which uses a small amount of training data, was explored to realize automatic segmentation of top-view images of a maize population at the seedling stage. Morphological and color related phenotypes were automatic extracted, including maize shoot coverage, circumscribed radius, aspect ratio, and plant azimuth plane angle. The results show that the approach can segment the shoots at the seedling stage from top-view images, obtained either from the UAV or tractor-based high-throughput phenotyping platform. The average segmentation accuracy, recall rate, and F1 score are 0.96, 0.98, and 0.97, respectively. The extracted phenotypes, including maize shoot coverage, circumscribed radius, aspect ratio, and plant azimuth plane angle, are highly correlated with manual measurements ($R^2$ = 0.96–0.99). This approach requires less training data and thus has better expansibility. It provides practical means for high-throughput phenotyping analysis of early growth stage crop populations.

## Introduction

Recently, plant phenotyping has become a rapidly developing data-intensive discipline [1,2]. Studying the phenotypic information of plants under different environmental conditions provides insight into plant genetics [3,4] and is important identifying and evaluating the phenotypic differences of different cultivars [5]. Field phenotypes are the manifestation of crop growth under real conditions and are an important basis for genetic screening and the

**Funding:** This work was supported by Construction of Collaborative Innovation Center of Beijing Academy of Agricultural and Forestry Sciences (KJCX201917), the National Natural Science Foundation of China (31871519, 32071891), Reform and Development Project of Beijing Academy of agricultural and Forestry Sciences, China Agriculture Research System (CARS-02), and the Construction of Scientific Research and Innovation Platform in Beijing Academy of Agriculture and Forestry Sciences (PT2020-24). HY is a paid employee of Beijing Shunxin Agricultural Science and Technology Co., Ltd. The funders had no role in study design, data collection and analysis, decision to publish, or preparation of the manuscript.

**Competing interests:** The authors have read the journal's policy and declare the following competing interests: HY is a paid employee of Beijing Shunxin Agricultural Science and Technology Co., Ltd. There are no patents, products in development or marketed products associated with this research to declare. This does not alter our adherence to PLOS ONE policies on sharing data and materials.

identification of mutations in field crops [6]. Therefore, it is important to conduct analyses of crop phenotypes under field conditions with high-precision. Traditionally, field phenotypic traits were obtained by manually measuring traits, which is work-intensive and time-consuming, limiting the number of measurable phenotypic traits. The development of information technology has made it possible to automatically acquire multi-source data of crops using high-throughput technology, such as images, point clouds, and spectrally collected data in the field, which can greatly reduce the manual labor and time commitment required to obtain crop phenotypic information. The cost of point cloud and spectral data acquisition sensors are more expensive than the image sensors; thus, image-based plant phenotyping has become a hot topic in agricultural research in recent years [7].

Unmanned aerial vehicles (UAVs), manned ground vehicles (MGVs), and tractor-based high-throughput phenotyping platforms (HTPPs) can rapidly obtain high-resolution top-view images of crop canopies. Researchers can extract phenotypic parameters [8], such as plant size [9], shape [10], and color [11], from the acquired images. For some specific phenotypic parameters, these approaches can be substituted for traditional manual measurements, improving the efficiency of collecting plant phenotypic information. However, different data collection methods and different environments can generate inconsistent image data. Thus, reliable automated methods are needed to extract accurate phenotypic information from large, complex datasets. Recently, researchers have proposed a variety of algorithms to address the above problems [12,13]; the basis of these algorithms is image segmentation.

Accurate and efficient field crop image segmentation methods can rapidly and accurately obtain crop phenotypic traits. Researchers have conducted numerous studies on the image segmentation of crops under field conditions [14,15]. Early field crop image segmentation methods can be roughly divided into four categories: shape constraints [16], edge detection [17], deep information integration [18], and machine learning methods [19]. These studies can address issues in the field, such as disease identification [20,21], environmental stress [22], chlorophyll diagnosis [23], and phenotypic extraction [24], at the individual plant or population scale. However, the background of the plant images in these methods was manually constructed or relatively simple. In addition, these methods typically have strict requirements on the light intensity of the input images.

With its powerful feature extraction capabilities, deep learning technology is a turning point for accurately and rapidly addressing image segmentation problems [25,26]. Fully trained models can achieve accurate image segmentation for regions of interest (ROI). Currently, popular deep neural network processing methods use center point detection [27]. Alternately, deep neural network processing methods directly perform leaf edge detection [28] to achieve image segmentation and whole or partial segmentation of images of plants collected under field [25] or indoor [31] conditions. Segmentation results are used to extract crop features [30], as well as quantify [29,31], count [11], and estimate diseases [32,33]. Deep learning has advantages in collaborative applications such as the interactions between genotype and environment. Compared with classical methods, deep learning technology does not rely on manual filters and feature annotations; instead, it learns the best representation of the data, allowing it to perform better in scenarios where the amount of data is sufficient.

Researchers have applied advanced hardware facilities and intelligent data processing methods to research plant phenotypes. However, accurate extraction of fine-scale phenotypic information of individual plants is still difficult under field conditions because of the occlusion and crossover that occurs in the later growth stages of crops. In a field maize population, for example, leaves of adjacent shoots appear cross-shaded after ridging, which makes it difficult to completely and precisely extract the phenotypes of individual plants within a population. Therefore, obtaining and resolving phenotypic traits of maize shoots at the seedling stage is a

better way to characterize the phenotypic traits of individual plants, and can guide the structural-functional analysis of maize populations in later growth stages. For example, phenotypic traits at the seedling stage provide reference information, such as growth position, direction, and growth potential, for each shoot within the population.

In this paper, a full convolutional neural network based end-to-end image segmentation approach of maize population at the seedling stage, named PlantU-net, is proposed. Using this approach, each shoot within the population image is precisely localized and the phenotypes are extracted. The approach is expected to provide technical support for image processing and high-throughput phenotype extraction from the top-view images acquired by UAVs and field phenotyping platforms.

## Materials and methods

### Data acquisition

Data for PlantU-net model training and inbred line population analysis were obtained at the experimental field, Beijing Academy of Agriculture and Forestry Sciences, Beijing, China (39˚ 56′ N, 116˚ 16′ E). A population used for correlation analysis with 502 cultivars [34] was planted in the field on May 17, 2019. The row and plant spacing were 60 and 27.8 cm, respectively. The planted cultivars can be divided into four subpopulations [35]: hard stalks (SS), non-hard stalks (NSS), tropical and subtropical (TST), and mixed inbred lines (Mixed), with 32, 139, 221, and 110 cultivars for each subpopulation. Top-view images of shoots were obtained 12 (V3) and 26 (V6) days after sowing. Images were acquired using an EOS5DIII digital camera with a 24–70 mm lens vertically downward mounted on a SLR tripod (height 1.7 m, as shown in Fig 1a), with each image containing approximately five to six plants. Size-known markers are placed in the original image to provide a scale reference for later image cropping and scaling. When acquiring the images, the experimenter faced to east, ensuring that the left side of the captured images was oriented to the north. Image acquisition occurred over three days (one day for V3; two days for V6): one sunny day for V3, and one sunny and one cloudy day for V6. The incident light angle and the intensity differed over the course of data acquisition. Drip irrigation belts were arranged to ensure adequate water and fertilizer. Consequently, these changes in the background cause challenges for later image processing. Top-view images of maize populations at the seedling stage were obtained using UAV and a field phenotyping platform.

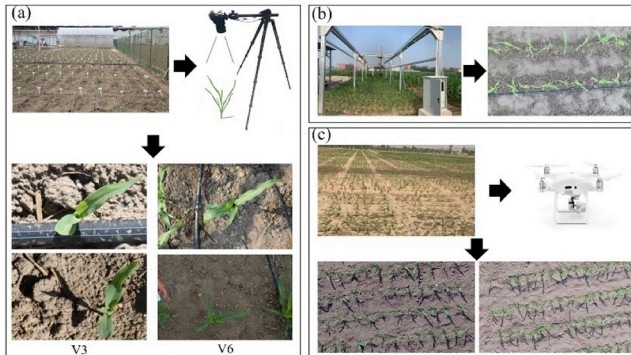

**Fig 1. The set up for acquiring the photographs and examples of acquired images.** This includes acquisition of top-view images using a tripod camera system (a), tractor-based high-throughput phenotyping platform (b), and UAV platform (c).

The experimental plots of the field phenotyping platform were adjacent to the plots obtained from the above-mentioned model dataset and managed in the same manner. Thirteen maize hybrids were planted within the coverage of the platform on May 25, 2019; this included one row of each hybrid, with 1.5 m long rows and 60 cm row spacing. The platform's image acquisition system consisted of a stable imaging chamber and a Hikari MV-CA060-10GC color camera. The camera lens was 2.5 m above the ground, and the resolution of the captured images was $3072 \times 2048$ pixels. In the process of data acquisition, the imaging chamber was equipped with a lens that moved in an S-shaped trajectory above the experimental plot, and the acquired images were stitched together to obtain a complete top-view image of the plot. The data acquisition of the tractor-based phenotyping platform is shown in Fig 1b. The image acquired on the 17th day after sowing was selected for subsequent phenotypic analysis.

The experiment of top-view image data acquisition using UAV of maize populations at the seedling stage was carried out at the Tongzhou Experimental Field, Beijing Academy of Agricultural and Forestry Sciences (39˚70′ N, 116˚68′ E). One hybrid of maize was grown on April 28, 2019 and planted in rows 2.1 m long and 60 cm apart. A visible light sensor was mounted on a UAV and image data was acquired 20 days after sowing. The image capture system consisted of a 1-inch CMOS HD camera and an engineering-specific gimbal. The UAV flew at an altitude of 30 m, and the resolution of the captured images was $4000 \times 3000$ pixels. The data acquisition process is shown in Fig 1c.

## Data preparation

Datasets with annotated images are necessary for robust image segmentation models. In practice, this dataset was constructed using the top-view images obtained at two periods, V3 and V6 (Fig 1a). Because the soil background accounts for a large proportion of the raw images, the images were cropped around the area containing the plants and the images were scaled to $256 \times 256$ pixels for further training the model. A total of 192 images, containing seedling maize shoots, were annotated using LabelMe software. Among the total number of images, 128 images were expanded into 512 images to use as a training set after mirror symmetry, translation, and rotation. The remaining 64 labeled images were used to form a validation set to determine the criteria that may prevent network training. To prevent overfitting, the network will train until the losses on the validation set are stable. The model designed in this study is a small sample learning model, and data augmentation was adopted to ensure the quality of the training set, which will be discussed later. There are 200 images in the testing set, which were randomly selected from the four maize subpopulations described in the experiment in Fig 1a. Here, images of 50 hybrids belonging to each subpopulation were randomly selected (subpopulation SS consisted of only 32 hybrids, so there are 18 duplicated hybrid images belonging to the SS subpopulation in the test set).

## PlantU-net segmentation network

To accurately segment maize shoots at the seedling stage in field conditions from the top-view image, the shoots were segmented as the foreground and output as a binary image. However, top-view images of field maize are relatively complex with stochastic background and uneven light conditions. Consequently, existing models are not satisfactory to extract pixel features. To address this issue, we built a PlantU-net segmentation network by adjusting the model structure and key functions of U-net [36], which improves the segmentation accuracy of images taken under a complex environment.

**Model structure.**   PlantU-net is a network designed for the segmentation of top-view images of crops grown in the field. A full convolution network is adopted to extract hierarchical features via an "end-to-end" process. As shown in Fig 2, the feature contraction path is composed of three layer downsampling modules, each module uses a 3 × 3 convolution to extract one row feature, and a 2 × 2 pooling operation to reduce the spatial dimensionality. Two convolution operations are conducted after downsampling to adjust the input size of the extended path. Corresponding to the contracted path, the extended path includes three layer upsampling modules. In each upsampling module, a 2 × 2 up sampling convolution is first performed to expand the spatial dimension. Then the upsampled results are fused with the low-level feature maps in the corresponding contracted path to connect contextual information across adjacent levels. Two convolution operations are performed during the upsampling process to reduce the feature dimension and facilitate feature fusion. After upsampling, a 1 × 1 convolution is performed as the full connection layer to output the segmented image. The same padding is filled in the samples during the convolution operations, which facilitates the computation. The parameters used for each layer of the model are shown in Table 1.

To a certain extent, the network parameters of the model are reduced to ease the burden of computers, and also to reduce the training time while ensuring the segmentation effect. Since the number of training samples is small, a dropout layer is appropriately added to prevent overfitting. In addition, to identify and utilize edge features, a maximum pooling layer is adopted for downsampling.

**Main functions.**   *Activation Function*. The activation function in deep learning incorporates nonlinear factors to solve the linear classification problem. In PlantU-net, Leaky ReLU is used as the activation function. It still has an output when the input is negative, which eliminates the neuron inactivation problem in back propagation. The expression is:

$$f(x) = \begin{cases} x \ \ if \ \ x \geq 0 \\ \theta x \ \ if \ \ x < 0 \end{cases} \tag{1}$$

For the final output layer of the model, Sigmoid is used as the activation function for biclass. Sigmoid is capable of mapping a real number to an interval of (0, 1), and is applicable for biclassing. Its expression is:

$$S(x) = \frac{1}{1 + e^{-x}} \tag{2}$$

*Loss Function*. The loss of function in the U-net model is replaced by the binary-cross-entropy function in the PlantU-net model. The binary-cross-entropy function is a cross-entropy of two-

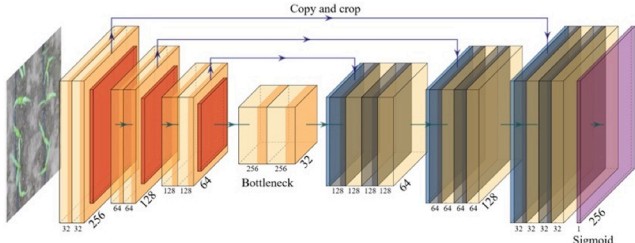

**Fig 2.  Architecture of the PlantU-net network.** The input is a 256×256×3 image. The hidden layer of the network includes downsampling (left) and upsampling stages (right). Both stages comprise convolution (Conv), activation (Leaky ReLU), and max pooling operations. The output is a 256 × 256 × 1 segmented image.

**Table 1. Configuration of the model structure parameters.** Refer to Fig 3 for the architecture of the PlantU-net network.

| Layers | Input | Convolution filter | Output |
|---|---|---|---|
| Downsampling module 1 | 256×256×3 | 3×3×32 | 128×128×32 |
| Downsampling module 2 | 128×128×32 | 3×3×64 | 64×64×64 |
| Downsampling module 3 | 64×64×64 | 3×3×128 | 32×32×128 |
| Convolution module | 32×32×128 | 3×3×256, 3×3×128 | 32×32×128 |
| Upsampling module 1 | 32×32×128 | 3×3×128, 3×3×64 | 64×64×64 |
| Upsampling module 2 | 64×64×64 | 3×3×64, 3×3×32 | 128×128×32 |
| Upsampling module 3 | 128×128×32 | 3×3×32 | 256×256×32 |
| Convolution 1×1 | 256×256×32 | 3×3×1 | 256×256×1 |

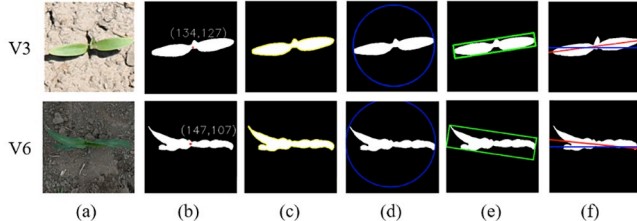

**Fig 3. Illustration of phenotype extraction based on the image segmentation results of V3 and V6 growth stages, respectively.** (a) The original image. (b) Coordination of the extracted center point. (c) Outline of the plant. (d) Minimum circumscribed circle. (e) Minimum bounding box. (f) Angle between the plant azimuth plane and the north. The red line represents the fitted azimuth plane and the blue line indicates the north–south direction. β is the angle between the red line and the blue line, and the value is between 0 and 180˚. The angle β between the red and blue lines was estimated and used to represent the angle of the plant azimuthal plane (Fig 3f).

class classifications, which is a special case of the entropy function. The binary classification is a logistic regression problem and the loss function of the logistic regression can also be applied. Considering the output of the last layer of the sigmoid function, this function is selected as the loss function. The mathematical expression of binary-cross-entropy function is:

$$L = -[y log y' + (1 - y) log(1 - y')] \tag{3}$$

where $y$ is the true value and $y'$ is an estimation when y = 1.

$$L = -log y' \tag{4}$$

The output of this loss of function is smaller when the estimated value is closer to 0, and the output value of the loss of function is larger when it is closer to 1. This is suitable for the binary classification output of the last layer in this network.

## Network training

The PlantU-net was trained using the Keras framework (Fig 1) with acceleration from GPUs (NVIDIA Quadro P6000). Five hundred and twelve images were used to train the model. Data expansion is the key to making the network have the required invariance and robustness because this model uses a small number of samples for training. For top-view images of maize shoots, PlantU-net needs to meet the robustness of plant morphology changes and value changes of gray images. Increasing the random elastic deformation of training samples is the key to training segmentation networks with a small number of labeled images. Therefore, during the data reading phase, PlantU-net uses a random displacement vector on the 3 × 3 grid to

generate a smooth deformation, where the displacement comes from a Gaussian distribution with a standard deviation of 10 pixels. Because the number of training samples is small, the dropout layer is added to prevent the network from overfitting. Through these "data enhancement" methods, the model performance is improved and overfitting is avoided. In each epoch, the batch size was 1, the initial learning rate was 0.0001, and adam is used as an optimizer to quickly converge the model. PlantU-net was trained until the model converged (the training loss was satisfied and remained nearly unchanged).

## Evaluation of segmentation accuracy

Because the segmentation of the top-view images of maize shoots using the PlantU-net model is considered a binary classification problem, when evaluating the segmentation results, the classification results of predicted output and ground truth (GT) data can be used to perform pixel-level comparisons. If the pixel in the leaves is marked as 1, and in the segmented image, the corresponding pixel is still 1, then it is judged as true positive (TP); if the pixel point is judged as 0 after segmentation, the pixel is judged as false positive (FP). Similarly, when the pixel in the original image does not belong to the maize leaf, it is marked 0, if such pixel is judged as 1 after segmentation, it is a false negative (FN); if such a pixel is also judged as 0, then it is a true negative (TN). Following these rules, three indicators for evaluation [37,38] were used in this study:

**(1) Precision.** Precision represents the proportion of true positive samples among those predicted to be positive and is defined as:

$$P = \frac{TP}{TP + FP} \tag{5}$$

**(2) Recall.** Recall indicates how many positive samples of the total sample are correctly predicted and is defined as:

$$R = \frac{TP}{TP + FN} \tag{6}$$

**(3) F1-Score.** After calculating the accuracy and recall, the F1-Score can be calculated, which represents the weighted harmonic average of accuracy and recall. It is used for standardized measurement and is defined as:

$$F1-Score = \frac{2PR}{P + R} \tag{7}$$

## Extraction of phenotypic parameters

The phenotypic traits concerning the shape and color characteristics of each shoot were estimated using PlantU-net based on the top-view images of the segmented maize shoots. The segmented images may still contain multiple maize plants. The phenotypic parameter extraction process will start with edge detection based on the segmentation results, connective domain markers based on the edge detection results, and finally single-plant phenotypic parameter extraction based on these connective domain markers.

**Morphological feature extraction.** The description of morphological features can be divided into two categories. The first category is the outline-based shape description, which focuses on describing the outline of the target area. The other category is the area-based shape description, which describes the target by area, geometric moment, eccentricity, and region shape. In this study, the center point Fig 3b) and contour (Fig 3c) of a maize shoot were first extracted from the segmented image. The minimum circumscribed radius (Fig 3d) and aspect ratio (Fig 3e) of the plant were then calculated based on the extracted contour. The coverage and plant azimuth plane were obtained based on the target region in the segmented images as described below.

1. The circumcircle radius (r) is half of the distance between the two pixels with the furthest outline of the plant (Fig 3d):

$$r = \frac{max[dis(C_i, C_j)]}{2} \tag{8}$$

where $C_i$, $C_j$ represent two pixels that are the furthest apart on the outline of the plant.

2. The aspect ratio (A) is the ratio of the length to width in the minimum bounding box of the plant (Fig 3e):

$$A = \frac{L}{H} \tag{9}$$

where L is the length in the x direction of the smallest bounding box and H is the length in the y direction. The smallest bounding box refers to the smallest rectangle among the n rectangles that can include the target plant area.

3. The segmented results are binary images; thus, the maize shoot coverage (C) is calculated by counting the total number of pixels occupied by the target area:

$$C = \sum_{x=0}^{m-1} \sum_{y=0}^{n-1} f(x, y) \tag{10}$$

where f (x, y) represents the binary map, m is the maximum number of pixels in the x-axis direction, and k is the maximum number of pixels in the y-axis direction. Regarding the binary maps, pixels of the target plant are always labeled by 1, whereas the background pixels are labeled using 0 for the output; therefore, the pixel method of calculation was used, meaning that pixels were counted as $f(x, y) = 1$ pixels. Calibration objects were used in the original image of the dataset. The length and width of the cropped image can be calculated using the calibration objects because the image size was cropped to $256 \times 256$. The area of each pixel was calculated according to the length and width of the image, and the size of the maize plant in the image was obtained by multiplying the total number of pixels in the segmented target area.

4. Studies have shown that expanded leaves of maize shoots are distributed along a vertical plane, which is the plant azimuth plane [39,40]. The original images for this study were oriented eastward during the data acquisition process; thus, the left side of the image in the dataset indicates the north. In the Fig 3f, the blue line indicates a single maize plant after segmentation and shows a north–south orientation. A red line was fitted by clustering in the leaf section (or tangent to it if the clustering result is a curve) as the plant azimuthal plane. The angle between the red line and the blue line was calculated as β, which was used as the azimuthal plane angle of the plant. The specific morphological features were extracted as shown in Fig 3.

**Extraction of color features.** Maize leaf color has a direct relationship with moisture, nutrients, and disease, so color characteristics are an important parameter for plant phenotyping [40]. The pixels in the image are composed of red (R), green (G), and blue (B) values. By aligning the segmentssed image as a region of interest (MASK) with the original image, the RGB parameters of the color features of the MASK region can be extracted, which can further be transformed into HSV color space parameters [41]. This approach was primarily used because the HSV model is similar to the color perception by the human eye, and the HSV model can reduce the effect of light intensity changes on color discrimination. Therefore, the parameters of the color phenotypes in this study are represented using the mean of the RGB or HSV parameters.

## Statistical analysis

The phenotypic traits extracted from segmentation results were compared with a manually measured value. The measured value of the circumcircle radius, aspect ratio, and plant azimuth plane was manually measured from the results by segmentation. Maize shoot coverage compared the segmentation results of PlantU-net with the results of manual segmentation. The adjusted coefficient of determination ($R^2$) and normalization root-mean-squared error (NRMSE) were calculated to assess the accuracy of these extracted parameters. The equations were as follows:

$$R^2 = 1 - \frac{\sum_{i=1}^{N}(vi - v'i)^2}{\sum_{i=1}^{n}(vi - \hat{v})i^2} \tag{11}$$

$$NRMSE = \frac{\sqrt{\frac{1}{n}\sum_{i=1}^{N}(v_i - v'_i)^2}}{\hat{v}}i \tag{12}$$

where n is the numbers of objects, $vi$ is the results of manual segmentation, $v'i$ is the value of PlantU-net, and $\hat{v}_i$ is the mean value of the results of manual segmentation.

In the phenotypic analysis of the four subpopulations, this study analyzed the phenotypic trait data extracted from the test set. Box plots were drawn using Python. The extracted phenotypic trait data was marked in Excel and Python was used to write a program to read the data. The data was then visualized by calling the Matplotlib development library in Python.

## Results

### Model segmentation effect

The PlantU-net segmented network has been trained many times. During the training process, each epoch contained 200 batches with the size of 1, and the final training loss was shown in Fig 4. Training losses declined quickly in the first 100 batches (Fig 4), and then became slower. The loss for the final partition is 0.003. The model was trained on a workstation (2 Intel Xeon (R) Gold 6148 CPU, 256 GB RAM and NVIDIA Quadro P6000 GPU) for 41 minutes.

To show the image segmentation of the PlantU-net model of a single maize plant, the dataset and training parameters used by the PlantU-net model were imported into the U-net model for training, and the segmentation results of the two methods were compared with the manual segmentation results (Fig 5). The segmentation of the PlantU-net model is better than that of the U-NET model. PlantU-net has a more complete edge detection of segmentation results, and the precision of pixel classification of the interested regions is higher.

Table 2 compares the training time and segmentation results obtained using the PlantU-net and U-net models. The segmentation precision of the PlantU-net model is significantly higher

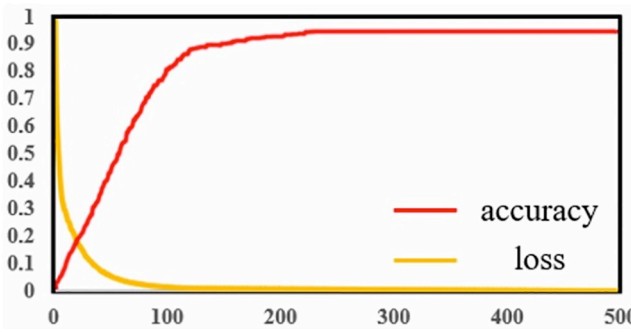

**Fig 4. Training loss curve within 500 epochs and prediction precision.**

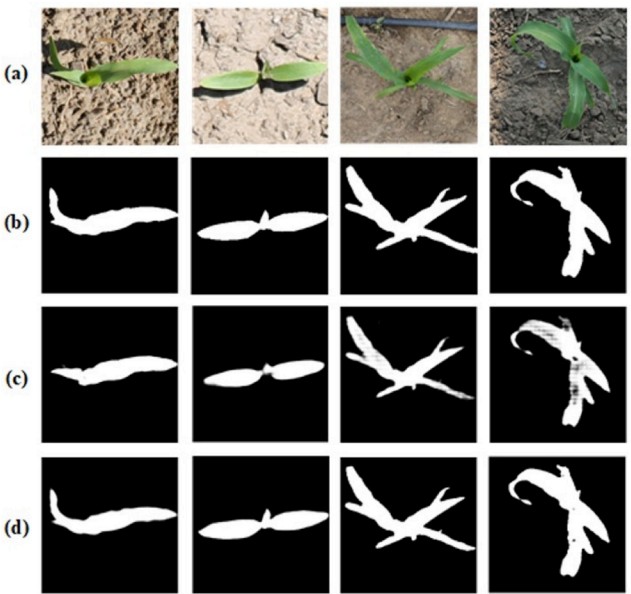

**Fig 5. Result of model segmentation, in which (a) is the original image, (b) is the ground truth by manual segmentation, (c) is the result obtained using the U-NET model, and (d) is the result obtained using the PlantU-NET model.**

**Table 2. Comparisons of the segmentation results of the U-net model and the PlantU-net model using the data obtained from the validation set (V) and test set (T).**

| Segmentation method | Training time/M | Precision | | Recall | | F1-Score | |
|---|---|---|---|---|---|---|---|
| | | V | T | V | T | V | T |
| U-net | 52 | 0.87 | 0.86 | 0.86 | 0.90 | 0.86 | 0.88 |
| PlantU-net | 41 | 0.95 | 0.96 | 0.98 | 0.98 | 0.96 | 0.97 |

than that of the U-net model, and less training time is required. In the segmentation results of the PlantU-net model, the values of the verification set and test set are similar. The PlantU-net model in the test set has a good segmentation effect, with the precision (P) of the segmentation results reaching 0.96, recall rate (R) reaching 0.98, and F1-score reaching 0.97.

To evaluate the segmentation performance of the proposed model on other plants, we compared the PlantU-net with orthogonal transform and deep convolutional neural network (OT-DCNN) [42], Mask R-CNN [43], and U-net [44] on the CVPPP benchmark datasets. Table 3 shows the comparison results, which demonstrates that the PlantU-net model performs better than the other three models, and is satisfactory for pot-grown plants.

## Results of single plant scale phenotypic parameter extraction

Using the PlantU-net model and the phenotype extraction method, the coverage, circumscribed radius, aspect ratio, and plant azimuth plane were determined using the validation dataset, and the measured data were compared with the extracted results for verification (Fig 6). Among them, the correlation coefficient $R^2$ of the artificial segmentation results and the automatic extraction results of the four morphological phenotypic parameters were all greater than 0.96, and the NRMSE values were all less than 10%, indicative of the reliability of the PlantU-net segmentation model and the phenotypic extraction method.

## Results of population scale phenotypic parameter extraction

To evaluate the performance of the PlantU-net model in the image segmentation and phenotypic parameter extraction of the maize population, the field high-throughput phenotypic platform and the top-view of maize seedlings obtained by UAV were selected as inputs. The top-view images were obtained using both the field orbital phenotypic platform and UAV. Fig 7 shows the segmentation results and schematic diagram of phenotypic parameter extraction of the PlantU-net model applied to two sample plots.

Phenotypic parameters were extracted from the segmentation results of two sample plots using the above methods. The mean value and standard deviation of various morphological parameters of the same cultivar of maize are shown in Table 4. The mean value can be used to quantify the growth potential of different maize cultivars in the same growth period, while the standard deviation can be used to evaluate the consistency of plant growth within the same maize cultivar. Therefore, this method can provide techniques for quantitative evaluation of plant growth potential, allowing for phenotypic analysis of the top-view of a maize population at the seedling stage obtained using multiple high-throughput phenotyping platforms in the field.

## Analysis of phenotypic differences among plant subpopulations

Phenotypic parameters were extracted from the images of the test set, and four phenotypic parameters, including coverage, the angle of plant azimuth plane, aspect ratio and circumscribed radius were statistically analyzed from the perspective of subgroups. Fig 8 shows the results from the phenotypic parameter analysis extracted from the image segmentation results of the test set. Among the four subgroups there were no statistical differences between the azimuth plane of plant growth and the included angle of due north (Fig 8b), while the other three phenotypic parameters all had differences within subgroups. In the analysis of the other three phenotypic parameters, the extracted values of SS and NSS subgroups were similar, which was

**Table 3. Segmentation comparison of PlantU-net model with three neural network-based approaches.**

|  | Precisioin | Recall | F1-Score |
|---|---|---|---|
| OT-DCNN [42] | 0.96 | 0.94 | 0.95 |
| MR-CNN [43] | 0.88 | - | - |
| U-net [44] | 0.96 | 0.74 | 0.83 |
| PlantU-net | 0.96 | 0.97 | 0.96 |

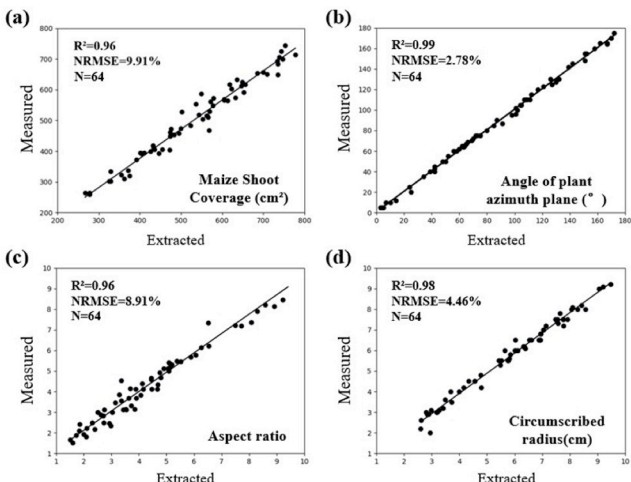

**Fig 6. Correlation analysis between phenotypic measurements and manual measurements in top-view segmentation results of field maize.** (a) Coverage, (b) the angle of plant azimuth plane, (c) aspect ratio, and (d) circumscribed radius.

related to the temperate zone of the two groups of cultivars. The TST subgroup includes tropical and subtropical cultivars, so the extracted parameters are different from the SS and NSS subgroups. However, the differences of the Mixed subgroup are relatively distinct. The results of the coverage analysis (Fig 8a) shows that the coverage value of the Mixed subgroup in the test set is low; in contrast, the results of the circumscribed radius (Fig 8d) showed a higher extracted value for the Mixed subgroup than that of the SS and NSS subgroups. This indicates that the leaves of the Mixed subgroup are more slender, resulting in low plant coverage and high leaf extension during the same growth period.

In terms of color phenotype, RGB and HSV phenotypic traits were extracted from the top image of the plant. Considering the segmented mask region is composed of many pixels, the mean value of the color of the pixels in the region is taken as the color phenotypic parameter of the plants. Similarly, based on the spatial color information of RGB and HSV, color traits of maize plants of different subgroups were analyzed (Fig 9). According to the analysis of RGB values, there was no obvious difference among the subgroups of all cultivars. In the analysis based on HSV color information, the TST and NSS subgroups did not show evident differences in color; however, the color difference between the TST and NSS subgroups was clear (the H and S of the cultivars in the NSS subgroup were higher than those in the TST subgroup). Approximately 1/3 of cultivars in both the SS subgroup and the Mixed subgroup were different from other cultivars in this subgroup (both H and S were higher than other cultivars in this subgroup).

The above results indicated that the PlantU-net model and phenotypic trait extraction method could be used to quantitatively analyze the morphological and color phenotypic trait differences among subgroups, which was suitable for a correlation analysis of genotype–phenotype.

## Discussion

### Image segmentation

At present, the threshold segmentation method is often used to segment top-view images of field crops. Although threshold segmentation with specific constraints can achieve very similar segmentation results [38,45], threshold segmentation is sensitive to noise and the effect on target segmentation is not ideal when there is little difference in gray scale. Threshold

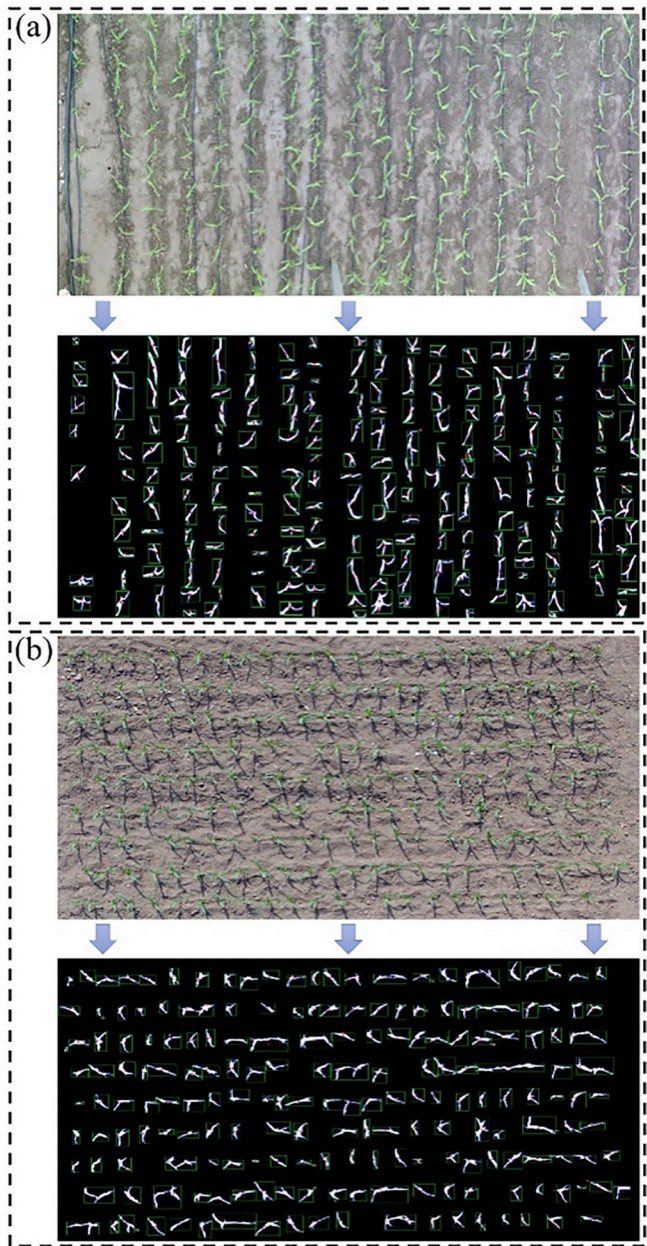

**Fig 7. Schematic diagram of the segmentation and phenotype extraction of the top-view of maize seedlings obtained using the field orbital phenotype platform and UAV.** In (a), a field track phenotypic platform was used to obtain the top-view images. In addition to the protected rows, a total of 13 varieties of maize plants were included (marked with a serial number in the fig). (b) For the top-view of the maize population obtained using the UAV, the image contains 216 maize plants of the same cultivar.

segmentation in different application scenarios (such as light and soil background) is relatively dependent on the selection of an empirical threshold. Manually setting different thresholds will greatly increase the workload of the interaction of the segmentation process, and it is difficult to achieve high-throughput in the processing of large quantities of data [46,47]. In comparison, this study designed the PlantU-net network model, which can implement end-to-end seedling stage of maize and group top-view as segmentation with the average segmentation

**Table 4. Morphological parameters of different maize cultivars.** AD268-M751 in the table corresponds to 1–13 in Fig 7a from top to bottom, and the bottom row of the data is obtained from the phenotypic parameters of maize plants in the image obtained by UAV.

| Cultivar | Coverage (cm²) | | Aspect ratio | | Circumscribed radius (cm) | | Angle of plant azimuth plane (°) | |
|---|---|---|---|---|---|---|---|---|
| | AVG | STD | AVG | STD | AVG | STD | AVG | STD |
| AD268 | 132.23 | 42.28 | 5.73 | 2.34 | 9.88 | 1.92 | 101.00 | 53.11 |
| MC670 | 150.44 | 45.41 | 4.13 | 2.08 | 10.73 | 2.21 | 85.30 | 45.68 |
| JNK2010 | 130.55 | 42.75 | 5.49 | 1.40 | 10.69 | 1.67 | 97.40 | 49.95 |
| JNK728 | 120.08 | 34.45 | 3.53 | 1.00 | 10.27 | 2.08 | 99.90 | 37.42 |
| NK815 | 117.45 | 50.69 | 5.30 | 2.60 | 9.03 | 1.64 | 87.90 | 55.44 |
| JKQC516 | 120.28 | 42.96 | 5.58 | 2.26 | 10.58 | 1.75 | 81.20 | 54.08 |
| SK567 | 119.87 | 53.66 | 5.54 | 2.58 | 9.23 | 1.29 | 74.90 | 51.28 |
| Y968 | 116.10 | 43.79 | 5.08 | 2.38 | 11.05 | 2.48 | 64.00 | 42.44 |
| MC141 | 119.04 | 38.42 | 6.07 | 2.01 | 10.79 | 1.71 | 101.90 | 59.28 |
| ZD958 | 103.31 | 45.97 | 6.17 | 1.94 | 11.13 | 2.16 | 94.90 | 53.55 |
| XY335 | 105.82 | 36.95 | 5.43 | 1.98 | 11.99 | 1.72 | 106.80 | 48.16 |
| JK968 | 112.15 | 45.98 | 4.86 | 2.44 | 10.28 | 2.08 | 87.40 | 51.96 |
| M751 | 150.84 | 44.48 | 4.77 | 2.69 | 10.63 | 2.42 | 92.30 | 52.82 |
| JNK728 | 115.92 | 37.41 | 5.35 | 2.08 | 7.97 | 1.47 | 89.57 | 49.42 |

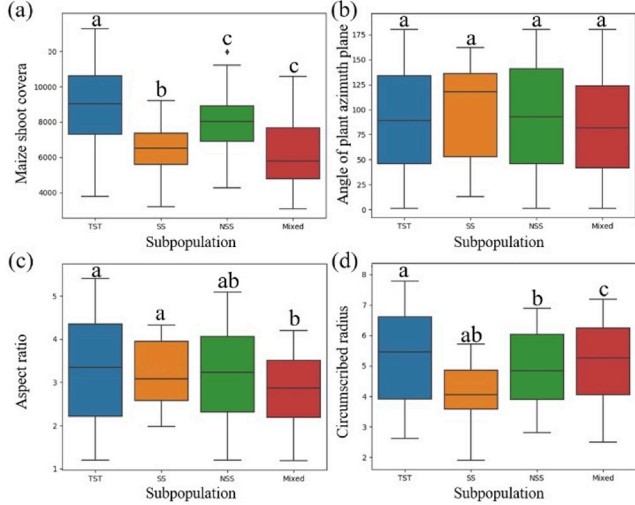

**Fig 8. Various phenotypic parameters were analyzed based on the differences of different subpopulations, in which the absence of shared letters indicated that the numerical differences of phenotypic parameters among subgroups were statistically significant (P<0.05).** (a) Coverage, (b) the angle of plant azimuth plane, (c) aspect ratio, (d) circumscribed radius.

precision of P = 0.96 and strong robustness. Under different light conditions and complex background features (the images used in this study have different background complexity, including weeds, drip irrigation, dry soil, and moist soil (Fig 1), different growth periods, and complex light environments (Fig 5a), accurate segmentation results were obtained without any human input. In addition, it only takes 0.04 s to extract various phenotypic parameters from the overhead image of a single maize plant (Fig 3) using PlantU-net. Moreover, it is only 0.6 s to extract phenotypic parameters from the top-view image of a population. The model can

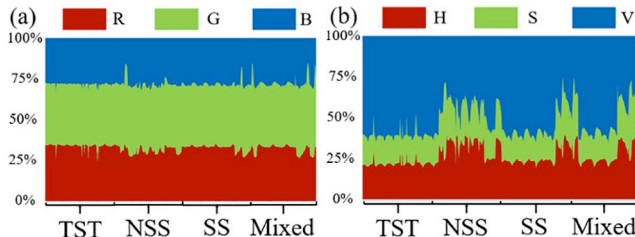

**Fig 9. Waterfall diagram of foreground plants in top-view images of maize plants of different subpopulations.** (a) RGB mean value analysis, (b) HSV mean value analysis.

achieve high-throughput phenotypic parameter extraction on the premise of ensuring segmentation precision.

Compared with other algorithms that use deep learning for image segmentation, the PlantU-net model can improve the segmentation precision by 10% compared with the U-net model [36] (Table 2), indicating that the PlantU-net model has higher credibility in the application of top-view images segmentation of maize plants at the seedling stage. The method proposed by Orsolya Dobos et al. [48] uses U-net and 2,850 images to train the Arabidopsis image segmentation model, while the PlantU-net model only needs 512 images for training and the training data does not need complex pre-processing, indicating that PlantU-net achieves high-precision segmentation with less training data. Therefore, when PlantU-net is used to solve image segmentation problems in other crops at the seedling stage, only a small number of annotated images are needed, indicating that the method is highly scalable. Yanan Li et al. [49] proposed a method called DeepCotton to deal with the segmentation of cotton in the field from coarse to fine. First, the fully convolutional neural network (FCN) was used for the end-to-end segmentation of self-collected field images. After extraction of network features, the "UP" algorithm is proposed to correct the defects in the image. This method sacrifices processing efficiency by ensuring segmentation precision; the processing time of this method is approximately 6 s, whereas using PlantU-net to segment a single image only requires approximately 0.6 s.

### Phenotypic analysis

Crop phenotype extraction based on data from top-view images is the main way to obtain phenotypes from high-throughput phenotyping platforms for many crops [50]. For example, Zhou et al. [38] extracted the phenotypic parameters of maize seedlings from the gray scale images collected by a UAV phenotyping platform through Otsu threshold segmentation and skeleton extraction methods. This method has a good segmentation effect on the overall image, but the extraction precision of the phenotypic traits of individual plants is limited. The correlation between the seedling emergence rate determined using the plant-bearing plane statistics and the measured data is only R = 0.77–0.86. In contrast, the PlantU-net segmentation network can not only segment the top-view image of a single maize plant at the seedling stage with high precision (Fig 5), but also extract phenotypic parameters with a higher correlation with measured data ($R^2 > 0.96$). The results show that the PlantU-net method can replace artificial measurement and threshold segmentation for quantitative extraction and evaluation of phenotypic traits.

The location and direction of the maize plant remains relatively unchanged, and the method overcomes the problem of the plants overlapping each other when viewed from above.

Therefore, the information of plant growth and plant azimuth-plane angle extracted from the top-view image of a maize population can provide measured data driving 3-D modeling of a maize population [51] and light distribution calculation and analysis [52] in the later growth stages. At present, the technology and equipment of high-throughput phenotyping platforms [53], including UAV [54], vehicle-based [55], and track-type, are developing rapidly, allowing for the collection of phenotypic data throughout the whole growth period. PlantU-net can also be applied to phenotypically analyze the top-view of a crop population obtained by multiple phenotypic platforms and can solve problems such as continuous monitoring of plant selection, analysis of plant growth difference between different plots, and analysis of plant growth consistency within the same treatment. These collected data would provide practical technical means for field crop breeding and cultivation research [56].

This study showed the applicability of the PlantU-net model in the extraction of phenotypic parameters in the seedling stage of maize. However, due to a large number of cross-shading in the top-view images caused by the overlapping of different plant leaves, this model could not solve the problem of phenotypic extraction in the middle and late stage of maize plant growth and development. Future work must determine how to use top-view continuity and the edge detection ability of the PlantU-net model to achieve the phenotypic extraction of plants in the middle and late stages of crop plants.

## Conclusion

In this study, an end-to-end segmentation method named PlantU-net was proposed based on the fully convolutional network, which improved the high-throughput segmentation performance of a top-view image of a seedling population and realized the accurate extraction of phenotypic data. The PlantU-net model had an average segmentation precision of 0.96 for the aerial image of maize plants at the seedling stage, and the phenotypic parameters extracted from the segmentation results were highly correlated with the values obtained by manual measurement ($R^2$ = 0.96–0.99). The model described in this manuscript is helpful for the segmentation of top-view images of the maize shoot, the extraction of phenotypes, and the quantitative evaluation of phenotypic traits obtained by high-throughput phenotypic platforms such as UAV and orbit.

## Supporting information

**S1 File. Original 128 images and corresponding labels of maize shoots for model training.** (ZIP)

**S2 File. Supporting data for Fig 6.** (ZIP)

**S3 File. Supporting data for Fig 8.** (XLSX)

**S4 File. Supporting data for Fig 9.** (XLSX)

## Author Contributions

**Data curation:** Yinglun Li, Haipeng Yan.

**Funding acquisition:** Weiliang Wen, Xinyu Guo, Chunjiang Zhao.

**Methodology:** Yinglun Li.

**Resources:** Xinyu Guo.

**Supervision:** Chunjiang Zhao.

**Validation:** Zetao Yu.

**Visualization:** Shenghao Gu.

**Writing – original draft:** Yinglun Li, Weiliang Wen.

**Writing – review & editing:** Yinglun Li, Weiliang Wen.

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
