## [Decision Letter · Decision Letter 0]

11 Nov 2020

PONE-D-20-32300

High-throughput phenotyping analysis of maize at the seedling stage using end-to-end segmentation network

PLOS ONE

Dear Dr. Wen,

Thank you very much for submitting your paper to PloS ONE. Your work has been evaluated by two experts in the field and there is consensus that it has merit.

You will see that the referees have suggested a number of changes that could be addressed in a minor round of revision. In particular:

 Please revise the reference format and adjust it to the journal style guide where needed; Consider complementing the related research with the references suggested by Reviewer #2.

We look forward to receiving your revised manuscript.

Kind regards,

Francesco Bianconi, Ph.D.

Academic Editor

PLOS ONE

Journal Requirements:

2. Our internal editors have looked over your manuscript and determined that it is within the scope of our Call for Papers on Plant Phenomics & Precision Agriculture. This collection of papers is headed by a team of Guest Editors for PLOS ONE. Additional information can be found on our announcement page: https://plos.io/phenomics.

If you would like your manuscript to be considered for this collection, please let us know in your cover letter and we will ensure that your paper is treated as if you were responding to this call. If you would prefer to remove your manuscript from collection consideration, please specify this in the cover letter.

Reviewers' comments:

Reviewer's Responses to Questions

**Comments to the Author**

1. Is the manuscript technically sound, and do the data support the conclusions?

Reviewer #1: Yes

Reviewer #2: Yes

2. Has the statistical analysis been performed appropriately and rigorously? 

Reviewer #1: Yes

Reviewer #2: Yes

3. Have the authors made all data underlying the findings in their manuscript fully available?

Reviewer #1: Yes

Reviewer #2: Yes

4. Is the manuscript presented in an intelligible fashion and written in standard English?

Reviewer #1: Yes

Reviewer #2: Yes

5. Review Comments to the Author

Reviewer #1: Dear Authors,

The subject of the study is interesting and topical, with high scientific and practical importance.

The introduction is presented correctly, in accordance with the subject. Numerous scientific articles, in concordance to the topic of the study, were consulted.

Methodology of the study was clearly presented, and appropriate to the proposed objectives.

The obtained results are important and have been analyzed and interpreted correctly, in accordance with the current methodology.

The discussions are appropriate, in the context of the results, and was conducted compared to other studies in the field.

The scientific literature, to which the reporting was made, is recent and representative in the field.

Some corrections have been suggested in the article.

1.

Page 26, Row 527:

“Most frequently in the article R2 was used; it is recommended to check, R or R2”

2.

References

The entire References chapter needs to be revised and corrected according to the Submission Guidelines, PLoS ONE Journal, References section.

In accord cu Submission Guidelines, PLoS ONE Journal, References section:

“References are listed at the end of the manuscript and numbered in the order that they appear in the text. In the text, cite the reference number in square brackets (e.g., “We used the techniques developed by our colleagues [19] to analyze the data”). PLOS uses the numbered citation (citation-sequence) method and first six authors, et al.”

Some corrections and examples were given in the article.

e.g

1.

“Zhao C, Zhang Y, Du J, Guo X, Wen W, Gu S, et al. Crop phenomics: Current status and perspectives. Front. Plant Sci. 2019;10:714. doi:10.3389/fpls.2019.00714.”

Instead of

“Zhao, C.; Zhang, Y.; Du, J.; Guo, X.; Wen, W.; Gu, S.; Wang, J.; Fan, J. Crop Phenomics: Current Status and Perspectives. Front. Plant Sci. 2019, 10, doi:10.3389/fpls.2019.00714.”

2.

“Yang W, Feng H, Zhang X, Zhang J, Doonan JH, Batchelor WD, et al. Crop Phenomics and high-throughput phenotyping: Past decades, current challenges, and future perspectives. Molecular Plant 2020;13:187-214. doi:10.1016/j.molp.2020.01.008.”

Instead of

“Yang, W.; Feng, H.; Zhang, X.; Zhang, J.; Doonan, J.H.; Batchelor, W.D.; Xiong, L.; Yan, J. Crop Phenomics and High-Throughput Phenotyping: Past Decades, Current Challenges, and Future Perspectives. Molecular Plant 2020, 13, 187-214, doi:10.1016/j.molp.2020.01.008.”

3.

“Holman F, Riche A, Michalski A, Castle M, Wooster M, Hawkesford M. High throughput field phenotyping of wheat plant height and growth rate in field plot trials using UAV based remote sensing. Remote Sens. 2016;8(12):1031. doi:10.3390/rs8121031.”

Instead of

“Holman, F.; Riche, A.; Michalski, A.; Castle, M.; Wooster, M.; Hawkesford, M. High Throughput Field Phenotyping of Wheat Plant Height and Growth Rate in Field Plot Trials Using UAV Based Remote Sensing. Remote Sensing 2016, 8, doi:10.3390/rs8121031.”

It is recommended to pay attention to the volume, number and pages (or number of the article) in the journal in which the article was published.

In accordance with the Submission Guidelines, PLoS ONE Journal, References section, it is not recommended that the all words be written in capital letters in the title of the article.

It is recommended to follow the instructions in the Submission Guidelines, PLoS ONE Journal, and References section.

Reviewer #2: This paper is very detailed concerning the aspects: the theory behind the proposed method, the dataset as well as the evaluation. The conclusions made from the experiments' results support the claimed contributions to the High-throughput phenotyping analysis of maize at the seedling stage. The paper is well written. But, the reviewer has some concerns regarding this paper which are given as follows.

1. It will be better to compare the proposed segmentation results with other recent segmentation methods and present it in the form of table.

2. It will be better if the recent papers in plant phenotyping are included as the references .

For eg: 1. Kumar, J. Praveen, and S. Dominic. "Rosette plant segmentation with leaf count using orthogonal transform and deep convolutional neural network." Machine Vision and Applications 31.1 (2020): 6.

2. Kuznichov, Dmitry, et al. "Data augmentation for leaf segmentation and counting tasks in Rosette plants." Proceedings of the IEEE Conference on Computer Vision and Pattern Recognition Workshops. 2019.

3. Kumar, J. Praveen, and S. Domnic. "Image Based Plant Phenotyping using Graph Based Method and Circular Hough Transform." J. Inf. Sci. Eng. 34.3 (2018): 671-686.

4. Sapoukhina, Natalia, et al. "Data augmentation from RGB to chlorophyll fluorescence imaging Application to leaf segmentation of Arabidopsis thaliana from top view images." Proceedings of the IEEE Conference on Computer Vision and Pattern Recognition Workshops. 2019.

6. PLOS authors have the option to publish the peer review history of their article (what does this mean?). If published, this will include your full peer review and any attached files.

Reviewer #1: No

Reviewer #2: No

---

## [Author Response · Author response to Decision Letter 0]

18 Nov 2020

Dear Editor,

 We would like to submit our revised manuscript, titled “High-throughput phenotyping analysis of maize at the seedling stage using end-to-end segmentation network”, for publication in PLOS ONE. A 'Revised Manuscript with Track Changes' version of this revised manuscript is uploaded. Detailed corrections and responds to the reviewer’s comments are as follows:

Reviewer #1s' comments:

1. Page 26, Row 527: Most frequently in the article R2 was used; it is recommended to check, R or R2?

Response: We checked the spelling in the article. R is used in reference [39], and is not a spelling error.

2. The entire References chapter needs to be revised and corrected according to the Submission Guidelines, PLoS ONE Journal, References section.

Response: We carefully checked and revised the format of the references.

Reviewer #2s' comments:

1. It will be better to compare the proposed segmentation results with other recent segmentation methods and present it in the form of table.

Response: Thanks for your comment. We added the comparison results of PlantU-net with other three recent segmentation methods according to the suggestions. For details, please see Table 3 and above contents in the revised version.

2. It will be better if the recent papers in plant phenotyping are included as the references.

Response: Thanks for your comment. We have cited these articles in the revised version.

Yinglun Li

College of Resources and Environment, Jilin Agricultural University, Changchun 130118, China; Tel: +86-10-5150-3362

Email: leeyinglun@163.com

13 Nov 2020

---

## [Editor Report · Decision Letter 1]

20 Nov 2020

PONE-D-20-32300R1

High-throughput phenotyping analysis of maize at the seedling stage using end-to-end segmentation network

PLOS ONE

Dear Dr. Wen,

Thank you for submitting your manuscript to PLOS ONE. After careful consideration, we feel that it has merit but does not fully meet PLOS ONE’s publication criteria as it currently stands. Therefore, we invite you to submit a revised version of the manuscript that addresses the points raised during the review process.

We look forward to receiving your revised manuscript.

Kind regards,

Francesco Bianconi, Ph.D.

Academic Editor

PLOS ONE

Additional Editor Comments (if provided):

There is one point that still needs further attention. The authors consider four figures of merit in the paper: precision, recall, F1-score (Eq. 7) and Sørensen-Dice coefficient (Eq. 8). I am unsure why  the latter two are treated as different parameters when they should actually be the same. It is essential to clarify.

---

## [Author Response · Author response to Decision Letter 1]

1 Dec 2020

Dear Editor：

Thank you for your valuable suggestions. Indeed, F1-Score and DICE (in Eq.7 and Eq.8) are two similar indicators. But in our opinion, they are different.

F1-score is an indicator to measure the accuracy of the two classification models. It takes account of the accuracy and recall of the classification model, which can be regarded as a weighted average of the accuracy and the recall. The Sørensen-Dice coefficient was usually used to estimate the similarity between the ground truth and the segmentation result. 

We found that these two parameters were also used in similar articles to evaluate the accuracy of the research results [43]. 

We add a short explanation at the end of this section in the manuscript: “F1-Score and DICE are two similar but different indicators. F1-Score can be regarded as the weighted average of the accuracy and the recall of the model, while DICE is used to evaluate the similarity between the GT and the segmentation result [43].”

Looking forward to your reply！

[43] Praveen K, Domnic S. Rosette plant segmentation with leaf count using orthogonal transform and deep convolutional neural network. Machine Vision and Applications, 2020, 31:6. doi: 10.1007/s00138-019-01056-2.

In addition, Financial disclosure and Competing interests have been supplemented as required.

---

## [Editor Report · Decision Letter 2]

17 Dec 2020

PONE-D-20-32300R2

High-throughput phenotyping analysis of maize at the seedling stage using end-to-end segmentation network

PLOS ONE

Dear Dr. Wen,

Thank you for submitting your manuscript to PLOS ONE. After careful consideration, we feel that it has merit but does not fully meet PLOS ONE’s publication criteria as it currently stands.

During the revision process a major methodological problem emerged – i.e., that the Sørensen-Dice Coefficient and the F1 score are treated as different metrics, whereas in fact they should give identical results in a one-class segmentation problem like the one investigated in the paper. This may be the consequence of a poor experimental design and/or a technical error somewhere in the computation pipeline. Consequently, a thorough revision is required before the manuscript can be given further consideration for possible publication in the journal.

We look forward to receiving your revised manuscript.

Kind regards,

Francesco Bianconi, Ph.D.

Academic Editor

PLOS ONE

Additional Editor Comments (if provided):

In a one-class segmentation task the Sorensen-Dice Coefficient and the F1 score are equivalent and should give the same result (please see the attached note.pdf).

---

## [Author Response · Author response to Decision Letter 2]

21 Dec 2020

Dear Editor：

Thank you for your valuable suggestions. 

After a detailed derivation, we found that the comment was indeed correct. The Sørensen-Dice Coefficient and the F1 score should give identical results. We have removed the duplicate evaluation metrics (The Sørensen-Dice Coefficient) in the revised version of this article. 

At last, thanks again for your valuable comments and patient guidance.

Best regards！

---

## [Editor Report · Decision Letter 3]

23 Dec 2020

High-throughput phenotyping analysis of maize at the seedling stage using end-to-end segmentation network

PONE-D-20-32300R3

Dear Dr. Wen,

We’re pleased to inform you that your manuscript has been judged scientifically suitable for publication and will be formally accepted for publication once it meets all outstanding technical requirements.

Kind regards,

Francesco Bianconi, Ph.D.

Academic Editor

PLOS ONE

---

## [Editor Report · Acceptance letter]

2 Jan 2021

PONE-D-20-32300R3 

High-throughput phenotyping analysis of maize at the seedling stage using end-to-end segmentation network 

Dear Dr. Wen:

I'm pleased to inform you that your manuscript has been deemed suitable for publication in PLOS ONE. Congratulations! Your manuscript is now with our production department. 

Kind regards, 

on behalf of

Prof. Francesco Bianconi 

Academic Editor

PLOS ONE